# Clinical, Functional, and Hemodynamic Profile of Schistosomiasis-Associated Pulmonary Arterial Hypertension Patients in Brazil: Systematic Review and Meta-Analysis

**DOI:** 10.3390/idr17020022

**Published:** 2025-03-04

**Authors:** Camila M. C. Loureiro, André L. Scheibler Filho, Vitor M. A. S. Menezes, Ricardo A. Correa, Rudolf K. F. Oliveira, Claudia Mickael, Joan F. Hilton, Brian B. Graham

**Affiliations:** 1Pulmonary Medicine, Santa Casa da Bahia, Salvador 40050-001, BA, Brazil; vitor.monti@outlook.com; 2Department of Medicine, Federal University of Bahia, Salvador 40170-110, BA, Brazil; 3Intensive Care Unit, Hospital Universitário Professor Edgard Santos, Salvador 40110-060, BA, Brazil; ascheibler@gmail.com; 4Department of Internal Medicine/Pulmonary Division, Federal University of Minas Gerais, Belo Horizonte 31270-901, MG, Brazil; racorrea9@gmail.com; 5Division of Respiratory Diseases, Department of Medicine, Federal University of Sao Paulo, Sao Paulo 04021-001, SP, Brazil; rudolf.oliveira@unifesp.br; 6Department of Medicine, University of Colorado, Aurora, CO 80045, USA; claudia.mickael@cuanschutz.edu; 7Department of Epidemiology and Biostatistics, University of California San Francisco, San Francisco, CA 94158, USA; joan.hilton@ucsf.edu; 8Department of Medicine, University of California San Francisco, San Francisco, CA 94143, USA; brian.graham@ucsf.edu

**Keywords:** schistosomiasis, pulmonary arterial hypertension, meta-analysis

## Abstract

Background: Schistosoma-associated pulmonary arterial hypertension (Sch-PAH), a complication of hepatosplenic schistosomiasis, is still underdiagnosed and undertreated. Sch-PAH is the third-most common cause of pulmonary arterial hypertension (PAH) in Brazil, and it is estimated that there are around 60,000 afflicted individuals. However, there is a lack of data on these patients, especially in endemic areas. Therefore, this study aimed to describe baseline demographic data, hemodynamic severity of disease, and functional impairment of Sch-PAH patients at diagnosis. Methods: For this systematic review, five databases (Embase, PubMed, SciELO, LILACS, and Cochrane) were searched to identify candidate publications reporting clinical, hemodynamic, and functional data at diagnosis of Sch-PAH patients referred to a PAH reference center in Brazil. Studies were excluded if they enrolled patients under the age of 18, the diagnosis was not confirmed by right heart catheterization (RHC), consisted of case reports, or did not report original data. Risk of bias was assessed using the Newcastle–Ottawa Scale and an adapted version for cross-sectional studies. Single-arm meta-analysis with a random-effect model was performed for each variable. Results: From 459 studies identified through systematic database searching, five studies were selected for this meta-analysis. The majority of the included patients were women (67%), New York Heart Association (NYHA) functional class III/IV (57%), mean age 49 years (95% confidence interval [95% CI], 46–52), 6 min walk distance 392 m (95% CI, 291–493), mean pulmonary arterial pressure (mPAP) 59 mmHg (95% CI, 56–61), pulmonary vascular resistance (PVR) 12 WU (95% CI, 11–13) and cardiac index (CI) 2.57 L/min/m^2^ (95% CI, 2.25–2.88). Conclusions: In summary, Sch-PAH has clinical characteristics similar to other forms of PAH, including connective tissue disease and idiopathic PAH. Additional studies or a unified registry would be essential for a better understanding of this relevant disease in Brazil.

## 1. Introduction

Schistosomiasis is a neglected tropical parasitic disease caused by infections with *Schistosoma* sp. worms. Despite its clinical importance, schistosomiasis continues to be a public health problem, especially in the developing world. An estimated 240 million people are infected and another 800 million are at risk of infection [1]. In 2020, schistosomiasis was endemic in 78 countries. More than 90% of people requiring preventive treatment with praziquantel live in Africa. Also, most deaths and disease-related disabilities occur in Africa [1].

In Brazil, the introduction of schistosomiasis occurred through the slave trade originating from the west coast of Africa. The disease initially spread in the Brazilian northeast and then to other areas, including Minas Gerais, due to population migration [2]. Currently, the disease is endemic mainly in northeastern and southeastern states. The greatest prevalence of schistosomiasis infection is in rural and economically disadvantaged areas where systematic testing and reporting is the lowest [3]. Thus, the 6 million estimated prevalence in Brazil may be an underestimate.

*S. mansoni*, transmitted by *Biomphalaria* snails, the only *Schistosoma* species that occurs endemically in Brazil, is particularly responsible for *Schistosoma*-associated pulmonary arterial hypertension (Sch-PAH). Although it is possible to develop the disease without evidence of portal hypertension [4], Sch-PAH is thought to be a complication that affects 5% to 10% of severe cases of hepatosplenic schistosomiasis, depending on the specific diagnostic modality and criteria [5]. Disease presentation includes symptoms of progressive right heart failure due to pulmonary hypertension, history of environmental exposure, infection or prior treatment for schistosomiasis, and evidence of hepatosplenic abnormalities mainly with periportal fibrosis [5,6].

Considering around 6 million people in Brazil are infected by *S. mansoni*, approximately 10%, or 600,000, may have hepatosplenic schistosomiasis [3,5], and thus there is likely to be at least 60,000 people with Sch-PAH in Brazil. Nevertheless, information is scarce regarding this population, especially in endemic areas. Formal diagnosis requires right heart catheterization (RHC), which is not widely available in resource-limited rural settings where the disease is most endemic. Additionally, awareness about this complication among general practitioners may be poor [7]. Thus, this systematic review of the literature and meta-analysis aims to better describe clinical, functional, and hemodynamic characteristics of Sch-PAH patients at the time of initial presentation and diagnosis.

## 2. Materials and Methods

### 2.1. Search Strategy and Selection Criteria

This systematic review and meta-analysis were undertaken in accordance with Meta-Analysis of Observational Studies in Epidemiology (MOOSE) guidelines and the Preferred Reporting Items for Systematic reviews and Meta-Analyses (PRISMA) statement [8,9], Appendix A. A systematic search of the literature was carried out on 28 October 2024 to identify candidate publications reporting clinical, hemodynamic, and functional data at diagnosis of Sch-PAH patients referred to a PAH reference center in Brazil. This survey included studies published from 1 January 2000 to 1 October 2024. All relevant studies were identified by cross-referencing the medical subject heading terms detailed in Appendix A and were retrieved from the following databases: PubMed, Excerpta Medica Database (Embase), Latin American and Caribbean Health Sciences Literature (LILACS), Scientific Electronic Library Online (SciELO), and Cochrane Library. Articles in English, Spanish, or Portuguese were considered.

Exclusion criteria were studies that enrolled patients under 18 years of age, PAH diagnosis based only on echocardiography and not confirmed by RHC (according to the hemodynamic definition at the time of the study i.e., mean pulmonary artery pressure (mPAP) > 25 mmHg and pulmonary artery wedge pressure (PAWP) ≤ 15 mmHg) [10], case reports, and studies that did not report original data (e.g., review articles) (Appendix A). For publications that constituted re-analyses of overlapping patient populations, we chose the publication that was deemed to provide the most pertinent information for each variable based on sample size.

The quality of included studies was assessed using the Newcastle–Ottawa Scale (NOS) for cohort studies [11] and an NOS version adapted for cross-sectional studies [12]. There were no unsatisfactory studies to be excluded. The study protocol was not previously published. 

### 2.2. Data Extraction

On full-text review, matched articles selected by two authors (C.M.C.L. and A.L.S.F.) were included for data extraction, while mismatched articles were resolved by consensus. One of the authors (A.L.S.F.) retrieved the data from all included studies, and a different author (C.M.C.L.) checked the extraction. Data extracted included study characteristics, patients’ demographics (age and gender), New York Heart Association functional class (NYHA-FC), 6 min walk distance (6MWD), hemodynamics (mPAP, PAWP, pulmonary vascular resistance (PVR), right atrial pressure (RAP), cardiac output (CO), cardiac index (CI) and mixed venous oxygen saturation (SvO2)) and laboratory data (brain natriuretic peptide (BNP) or N-terminal prohormone of brain natriuretic peptide (NT-proBNP)) from all included studies. There were two sets of overlapping samples selected due to complementary data (patients from the same time period at the same Brazilian reference center). One article in one of these sets presented only data categorized into two independent cohorts of patients (before 2010 and after 2010), the earlier one being overlapped. These cohorts were treated as independent studies to minimize bias [13]. For each variable, the authors analyzed the sets of overlapping samples to select the data with the greater sample size.

### 2.3. Data Synthesis

We used R Software version 4.4.2 (RStudio Software version 2024.09.1+394 with “meta” and “metafor” packages, Posit PBC, Vienna, Austria) to perform a single-arm meta-analysis with a random-effect model, in order to accommodate expected high between-study variances. Heterogeneity was assessed visually with forest plots and is presented with both I2 and τ2. Numerical data are presented as means and 95% confidence intervals (95% CIs), and qualitative data as proportions and 95% CIs. NYHA-FC was organized in two categories (FC I/II vs. FC III/IV), due to similar grouping among some of the studies.

## 3. Results

The search strategy yielded 459 potentially relevant publications (Figure 1). After titles were de-duplicated and abstracts screened, 39 records were retrieved for full-text review. The observed agreement between reviewers for eligibility of articles on the initial screening was 100%. Thirty-four publications were excluded for prespecified reasons (Figure 1, Appendix A). Overall, five studies were eligible for final inclusion, and their quality was judged by both reviewers completing the NOS checklist (Table 1).

The number of Sch-PAH patients in each study ranged from 92 to 198 individuals. BNP, NT-proBNP, and SvO2 appeared in only one sample and therefore were not analyzed. The mean age was 49 years (95% CI, 46–52), and 67% of patients were female (95% CI, 60–73%). The mean 6MWD was 392 m (95% CI, 291–493). The majority of the included patients were in NYHA-FC III/IV (57%, 95% CI, 49–64) (Figure 2).

Hemodynamic data of Sch-PAH patients at the time of diagnosis are shown in Figure 3. Mean mPAP was 59 mmHg (95% CI, 56–61), PAWP was 12 mmHg (95% CI, 11–14), PVR was 12 WU (95% CI, 11–13), RAP was 11 mmHg (95% CI, 10–12) and CI was 2.57 L/min/m^2^ (95% CI, 2.25–2.88).

## 4. Discussion

This meta-analysis assessed Sch-PAH characteristics in Brazilian patients. According to the results, Sch-PAH typically affects middle-aged females (2:1). Although presenting with a 6MWD within the intermediate-risk stratification according to current pulmonary hypertension guidelines [18], the fact that over half were NYHA-FC III/IV suggests the majority were highly symptomatic at diagnosis. These observations are similar to characteristics of other PAH etiologies; for example, the REVEAL registry found among idiopathic PAH (iPAH) patients a 4:1 ratio of female to male subjects, a mean 6MWD of 374 m, and 55% in NYHA-FC III/IV [19]. In this meta-analysis, baseline hemodynamic assessment revealed high PVR and RAP values, but CI within the low-risk stratification according to current recommendations. The REVEAL iPAH group had a mean mPAP of 52.1 mmHg and a CI of 2.2 L/min/m^2^ [19].

Despite its high worldwide prevalence, Sch-PAH is relatively understudied. In a series from one PAH reference center in Sao Paulo, Brazil, Sch-PAH was the third-most prevalent PAH subgroup [20]. However, Sch-PAH has not been significantly studied in other Latin American countries or registries worldwide [21].

Several series, which were included in our analysis here, that directly compared the hemodynamics between Sch-PAH and iPAH found that Sch-PAH patients generally have a more favorable hemodynamic profile, with lower PVR and higher CO than iPAH patients [14,20,22]. A previous systematic review of the literature also demonstrated a better hemodynamic profile and survival of Sch-PAH patients in comparison to iPAH patients from 11 clinical registries [23]. However, these findings may be confounded by sample overlap, since most of the included studies were derived from a single center in a non-endemic area (Sao Paulo), whereas in our meta-analysis, sample overlapping was avoided by study design.

In contrast, recent data from an endemic area of Brazil (Pernambuco) suggested that Sch-PAH actually has a comparable hemodynamic profile and survival to other PAH etiologies [17,24]. Furthermore, pulmonary artery enlargement has been observed to be more pronounced in Sch-PAH patients [22,25]. These data may be related to prolonged and continuous exposure to the parasite in individuals residing in endemic areas, with possible recurrent reinfection and/or increased susceptibility to infection by *S. mansoni* [26], whereas there may be referral bias in those presenting to a reference center in a non-endemic setting.

One of the limitations of any meta-analysis is that the level of evidence generated by the meta-analysis depends on the quality of the selected studies. In this meta-analysis, all the selected publications were observational studies. Another limitation is the possibility of selection bias, since the manuscripts reviewed were mostly from Brazilian PAH reference centers located in non-endemic areas for schistosomiasis. Sensitivity tests were not conducted because of the limited number of selected articles.

In summary, Sch-PAH has clinical characteristics similar to other forms of PAH, including iPAH. Therefore, we recommend that patients undergoing PH investigation should be evaluated for epidemiological and ultrasonographic data suggestive of schistosomiasis etiology. Also, asymptomatic individuals with hepatosplenic disease should be regularly monitored for cardiopulmonary symptoms and receive echocardiographic screening, especially in endemic areas. Since most of the data found on Sch-PAH patients in the medical literature come from a few reference centers in Brazil, some located in non-endemic areas, a unified Brazilian registry would help us understand the natural history of Sch-PAH and enable rigorous comparison to other PAH etiologies. This could additionally improve public health programs in identifying at-risk people who could potentially benefit from early diagnosis and specific therapy.

## Figures and Tables

**Figure 1 idr-17-00022-f001:**
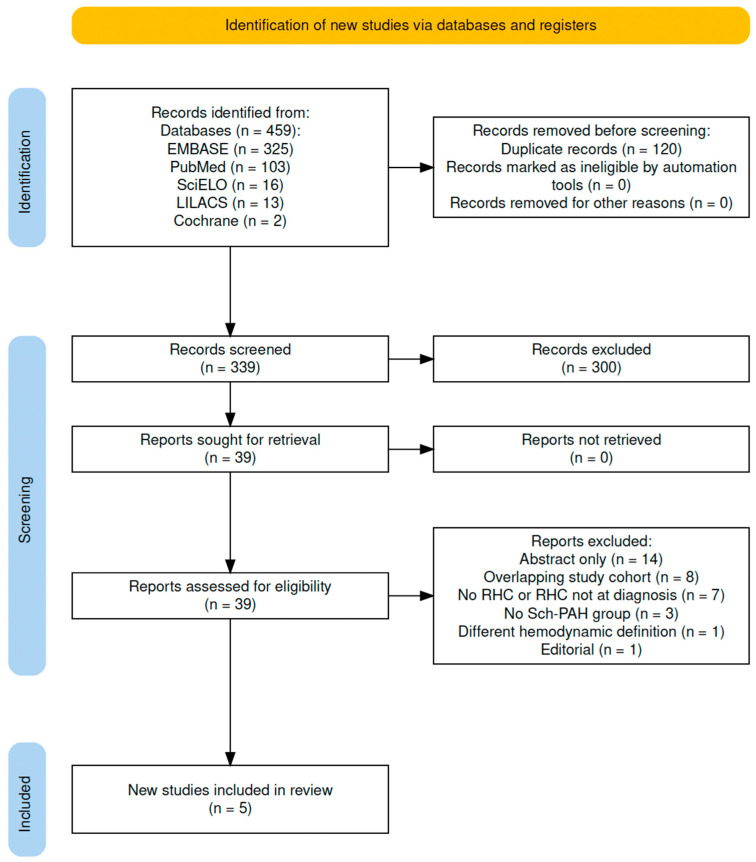
Flowchart of the search strategy and selection of the trials. Embase: Excerpta Medica Database; SciELO (Scientific Electronic Library Online); LILACS: Latin American and Caribbean Health Sciences Literature; RHC: right heart catheterization; Sch-PAH: Schistosoma-associated pulmonary arterial hypertension.

**Figure 2 idr-17-00022-f002:**
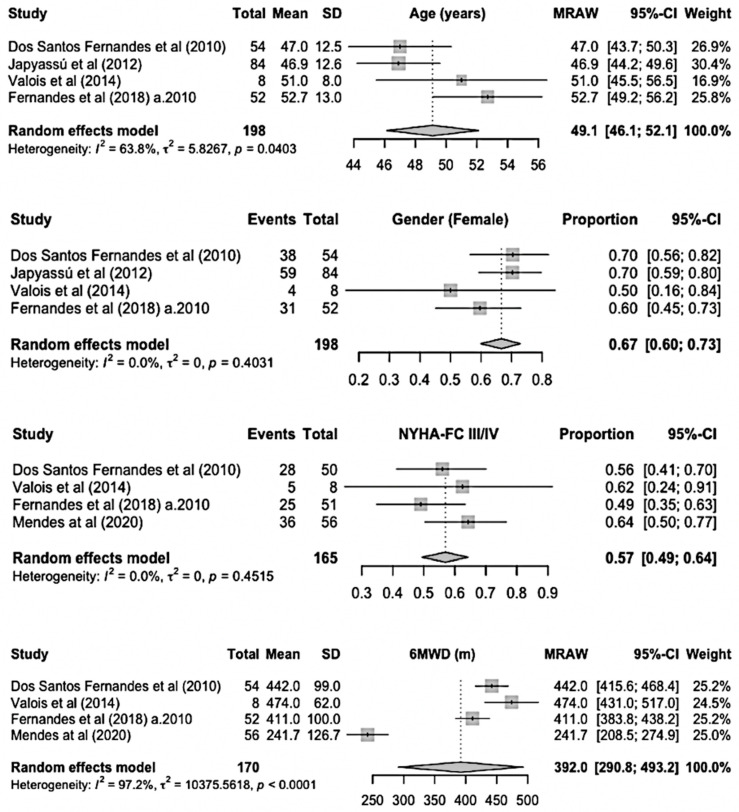
Baseline demographic data and functional impairment of Sch-PAH patients. a.2010: after 2010; NYHA-FC: New York Heart Association functional class; 6MWD: 6 min walk distance [13,14,15,16,17].

**Figure 3 idr-17-00022-f003:**
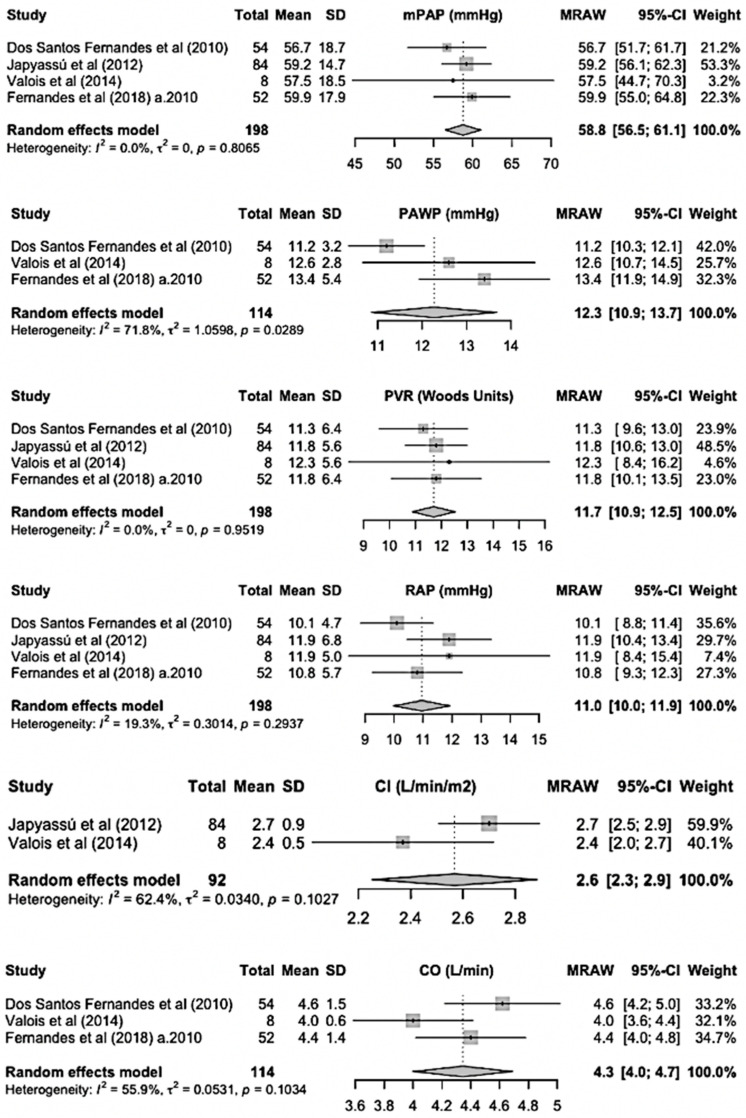
Baseline hemodynamic severity of disease of Sch-PAH patients. a.2010: after 2010; mPAP: mean pulmonary artery pressure; PAWP: pulmonary artery wedge pressure; PVR: pulmonary vascular resistance; RAP: right atrial pressure; CI: cardiac index; CO: cardiac output [13,14,15,16,17].

**Table 1 idr-17-00022-t001:** Included studies characteristics and risk of bias according to the Newcastle–Ottawa Scale.

First Author/Year	Study Design	Number of Sch-PAH	Study Period	Study Characteristics	Risk of Bias *
Dos Santos Fernandes et al. (2010) [14]	Retrospective cohort	54	2004–2008	Compared Sch-PAH to iPAH survival, demographic, functional, treatment and hemodynamic data	Good quality
Japyassú et al. (2012) [15]	Cross-sectional	84	2005–2009	Described demographic and hemodynamic data and correlated to NYHA-FC and 6MWD of Sch-PAH patients	Satisfactory
Valois et al. (2014) [16]	Cross-sectional	8	2007–2010	Compared Sch-PAH to iPAH using demographic, clinical and hemodynamic data and CPET variables	Very good quality
Fernandes et al. (2018) [13]	Retrospective cohort	102	2010–2018	Compared survival between treated and untreated Sch-PAH patients and presented demographic and hemodynamic data, NYHA-FC, 6MWD, BNP	Fair quality
Mendes et al. (2020) [17]	Cross-sectional	56	2001–2009	Compared Sch-PAH to non-Sch-PAH using demographic, clinical, hemodynamic and echocardiographic data, NT-proBNP	Very good quality

Sch-PAH: schistosoma-associated pulmonary arterial hypertension; iPAH: idiopathic pulmonary arterial hypertension; NYHA-FC: New York Heart Association functional class; 6MWD: 6 min walk distance; CPET: cardiopulmonary exercise test; BNP: brain natriuretic peptide; NT-proBNP: N-terminal prohormone of brain natriuretic peptide. * Risk of bias by Newcastle–Ottawa Scale (NOS) quality assessment and NOS adapted version for cross-sectional studies [11,12].

## Data Availability

All information necessary to replicate our results, including search strategy, are available on Appendix A.

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
