# Peer review of "Clinical, Functional, and Hemodynamic Profile of Schistosomiasis-Associated Pulmonary Arterial Hypertension Patients in Brazil: Systematic Review and Meta-Analysis"

_2036-7449, 2025, doi:10.3390/idr17020022_

Round 1

Reviewer 1 Report

Comments and Suggestions for Authors

Loureiro et al. review the occurrence of Schistosoma-associated pulmonary arterial hypertension (Sch-PAH) among a subpopulation with hepatosplenic schistosomiasis. They aim to provide specific diagnostic and clinical criteria for  Sch-PAH.

Line 17 "...around 60.000 afflicted individuals."

Lines 45-46: Rephrase for clarity.

Line 57: "...10% of which, or 600.000, ..."

Line 58: "...at least 60.000 people with Sch-PAH in Brazil."

1. Provide the characteristics of the five selected studies with regards to any descriptions of markers or pathophysiological pathways associated with Sch-PAH in the discussion section. Summarize the characteristics in a table or graphic.

2. What recommendations can the authors provide based on the results of the meta analysis, that would help public health programs and help identify at-risk individuals? Revise lines 191-197 to include the criteria that can aid the clinical identification of Sch-PAH. 

3. Include in the discussion section how the results of the meta analysis compares with the incidence and prevalence of Sch-PAH in other endemic areas around the globe and the basis (or criteria) for identifying at-risk individuals in other regions of the globe.

Author Response

Dear reviewer,

Thank you very much for revising our manuscript and for your input.

Regarding the suggestions made before numbers 1-3, we have rewritten for clarity.

In regard to the other suggestions, the answers are listed below:

1. Thank you for the question. None of the selected studies provided description of markers or pathophysiological pathways associated with Sch-PAH.

- Dos Santos Fernandes et al, 2010 - The authors evaluated demographic data, functional class, 6MWT, treatment, and performed survival analysis of patients with iPAH and Sch-PAH.

- Japyassu et al, 2012 - The authors compared hemodynamic data with functional class and distance in the six-minute walk test (6MWT) of Sch-PAH patients

- Valois et al, 2014 - The authors compared demographic data, hemodynamic parameters and cardiopulmonary exercise test variables between individuals with iPAH and ScHPAH.

- Fernandes et al, 2018 - The authors collected baseline clinical, demographical, functional class, 6MWT, BNP and haemodynamic data and analyzed survival rate of newly diagnosed Sch-PAH patients treated with PAH therapies against a group of untreated patients from a historical cohort.

- Mendes et al, 2020 - The authors extracted demographic, clinical, laboratory, echocardiographic, 6MWT, and hemodynamic variables to compare Sch-PAH and non-Sch-PAH patients.

2. We have rewritten the final paragraph.

“In summary, Sch-PAH has clinical characteristics similar to other forms of PAH, including iPAH. Therefore, we recommend that patients undergoing PH investigation should be evaluated for epidemiological and ultrasonographic data suggestive of schistosomiasis etiology. Also, asymptomatic individuals with hepatosplenic disease should be regularly monitored for cardiopulmonary symptoms and perform screening echocardiography, especially in endemic areas. Since most of the data found on Sch-PAH patients in the medical literature come from a few reference centers in Brazil, some located in non-endemic areas, a unified Brazilian registry would help understand the natural history of Sch-PAH and enable rigorous comparison to other PAH etiologies. This could additionally improve public health programs in identifying at-risk people who could potentially benefit from early diagnosis and specific therapy.”

3. In the introduction section, we estimated that Sch-PAH should afflict around 60,000 individuals in Brazil based on previous published data (references 3 and 5). We did not assess the incidence and prevalence of Sch-PAH in Brazil.

We attached a revised version of the manuscript for your appreciation.

Sincerely,

Camila Loureiro.

Reviewer 2 Report

Comments and Suggestions for Authors

1)  The present content of the MS will be helpful for sustaining public health.

2)   However, the conclusion is not easy to understand from the present MS.  They had better adding that to the "Discussion" part.   

3)   I have a quesstion.  Their appendixes with another word file will be added to the MS when published?   My ooppion is that absolutely no neded.  And, their figures 2 and 3 seem to be too long.   Thay should diminish them.      

Author Response

Dear reviewer,

1)Thank you very much for revising our manuscript and for your input.

2) Regarding your suggestion, we have rewritten the final paragraph for clarity.

“In summary, Sch-PAH has clinical characteristics similar to other forms of PAH, including iPAH. Therefore, we recommend that patients undergoing PH investigation should be evaluated for epidemiological and ultrasonographic data suggestive of schistosomiasis etiology. Also, asymptomatic individuals with hepatosplenic disease should be regularly monitored for cardiopulmonary symptoms and perform screening echocardiography, especially in endemic areas. Since most of the data found on Sch-PAH patients in the medical literature come from a few reference centers in Brazil, some located in non-endemic areas, a unified Brazilian registry would help understand the natural history of Sch-PAH and enable rigorous comparison to other PAH etiologies. This could additionally improve public health programs in identifying at-risk people who could potentially benefit from early diagnosis and specific therapy.”

3)The appendices will be published in a separate file as supplementary material, in order to fulfill MOOSE's and PRISMA's criteria. I discussed this with the other authors, and we feel that all the information should be preserved in both figures, providing a full comprehension of the results.

We attached a revised version of the manuscript for your appreciation.

Sincerely,

Camila Loureiro.

Round 2

Reviewer 1 Report

Comments and Suggestions for Authors

The authors have responded to the queries.

The features of the five studies as shown in the response to #1 should be added to Table 1. The features of the studies help provide clarification for why the studies were selected.

Author Response

Thank you very much for your suggestion. In accordance to it, we have changed Table 1.

Best regards.